



# An analogue based forecasting system for Mediterranean marine litter concentration

Gabriel Jordà[1,2, *] and Javier Soto-Navarro[3, *]

[1]Centre Oceanogràfic de les Balears, Spanish Institute of Oceanography (CN-IEO/CSIC). Mallorca, 07015, Spain.

[2]University of the Balearic Islands (UIB). Mallorca, 07122, Spain.

[3]Physical Oceanography Group of the University of Málaga (GOFIMA). Málaga, 29071, Spain.

*These authors contributed equally to this work

*Correspondence to:* Javier Soto-Navarro (javiersoto@uma.es)

## Abstract

In this work we explore the performance of a statistical forecasting system for marine litter (ML) concentration in the Mediterranean Sea. In particular, we assess the potential skills of a system based on the analogues method. The system uses a historical database of ML concentration simulated by a high resolution realistic model and is trained to identify meteorological situations in the past that are similar to the forecasted ones. Then, the corresponding ML concentrations of the past analog days are used to construct the ML concentration forecast. Due to the scarcity of observations, the forecasting system has been validated against a synthetic reality (i.e. the outputs from a ML modelling system). Different approaches have been tested to refine the system and the results show that using integral definitions for the similarity function, based on the history of the meteorological situation, improves the system performance. We also find that the system accuracy depends on the region of application being better for larger regions. Also, the method performs well to capture the spatial patterns but performs worse to capture the temporal variability, specially the extreme values. Despite the inherent limitations of using a synthetic reality to validate the system, the results are promising and the approach has potential to become a suitable cost effective forecasting method for ML concentration.

## 1. Introduction

The ubiquity of the plastic waste pollution in seas and oceans worldwide raises great concern in the society and the scientific community, as it poses a significant environmental and socioeconomic threat (UNEP, 2009). In consequence, the analysis of the impacts of marine litter (ML) pollution on the marine life and ecosystems has become a hot topic on marine research in recent years (Maximenko et al., 2019; Van Sebille et al., 2020; Lebreton et al., 2019; Lebreton and Andrady, 2019; Soto-Navarro et al., 2021). ML particles accumulate both in shallow and deep





waters, and particularly in enclosed basins such as the Mediterranean Sea (Soto-Navarro et al., 2020; Cózar et al., 2015), where the observed concentrations are in the same range of those measured in the great plastic patches formed in the subtropical gyres of the open oceans (Cózar et al., 2015; Law et al., 2014; Van Sebille et al., 2015). Moreover, risk analyses have shown that marine organisms in the Mediterranean basin can be highly impacted by ML pollution (Compa et al., 2019; Soto-Navarro et al., 2021). The starting point to analyze those impacts and to establish suitable mitigation strategies is to understand the spatial distribution and temporal evolution of the ML particles. Unfortunately, to carry on that analysis solely based on observations is not feasible. The large spatial and temporal heterogeneities of the field campaigns, along with the lack of standardized observational protocols, do not allow a synoptic representation of the ML distribution (see Maximenko et al. (2019) for a thorough analysis of the ML observations problems and proposed improvements). For these reasons, numerical modeling emerges as a fundamental tool to achieve a synoptic description of ML dispersion patterns and as the base for the forecasting systems that would reproduce its spatial variability and time evolution.

ML forecasting systems are usually based on the combination of two different numerical models (Lebreton et al., 2012; Van Sebille et al., 2015; Maximenko et al., 2012). On the one hand, an ocean circulation forecasting system is implemented to provide ocean currents. On the other hand, a lagrangian model uses those currents to simulate the advection and diffusion of passive particles in the ocean that mimic the evolution of ML. In the Mediterranean, several studies using this methodology have been carried out using current fields from high resolution regional models covering the whole basin (Liubartseva et al., 2018; Macias et al., 2019; Mansui et al., 2015; Soto-Navarro et al., 2020 ) or specific regions such as the Adriatic, the Tyrrhenian or the Aegean (Politikos et al., 2017; Fossi et al., 2017; Liubartseva et al., 2016; Palatinus et al., 2019). This modelling approach is considered to be the most accurate choice for ML forecasting (Van Sebille et al., 2020) provided the ML inputs are correctly prescribed (Liubartseva et al. (2018) , Soto-Navarro et al. (2020)).

The downside of developing a forecasting system based on the direct modelling approach is that it involves a high technical complexity and computational cost. In order to overcome this limitations, it might be possible to develop a fast and light forecasting system based on statistical methods. One choice would be the so called Statistical Downscaling Methods (SDMs) which relies on determining statistical relationships between large scale variables (usually atmospheric patterns) and local variables. They are broadly used in atmospheric modelling to forecast the evolution of local variables from large scale atmospheric models. The advantage of the SDMs is that the mathematical relationship derived by the model between the local and the large scale variables is valid not only for the present climate, but can also be used to estimate the future evolution of the local variables. In summary, the SDMs provide a simplified 'static' methodology



to forecast the evolution of local variables without the need of running a complex dynamical models. There are numerous downscaling methodologies based on different statistical properties. Among them, the analog method (Lorenz, 1969) is the most broadly used due to its simplicity and accuracy (Grouillet et al., 2016). This technique assumes that similar (or analog) atmospheric patterns over a given region, represented by large scale atmospheric variables or predictors, lead to similar local meteorological outcomes (or predictands) in a particular location. This assumption provides a simple algorithm to downscale the local occurrence of the variable of interest from a given large scale atmospheric pattern (see section 2.1 for a detailed description). In general, it has been shown that the analog method performs as well as other more sophisticated downscaling techniques (Zorita and von Storch, 1999), indicating that it is an efficient alternative for many downscaling problems. Its main advantages are that is non-parametric (i.e. no assumptions are made about the distribution of the variables used as predictors), non-linear (i.e. it can take into account the non-linearity of the relationships between predictors and predictands), and it is spatially coherent (i.e., preserves the spatial covariance structure of the local variables). The analog method has been satisfactorily applied in the Mediterranean region not only for the downscaling of meteorological or hydrological variables such as precipitation or river runoff (Grouillet et al., 2016; Wu et al., 2012; Caillouet et al., 2016), but also for the reconstruction of sea surface temperature in the glacial period (Hayes et al., 2005), the assimilation of satellite derived sea surface height (Lopez-Radcenco et al., 2019) and the projection of complex climatic impact indices such as the fire weather index or the physiological equivalent temperature (Casanueva et al., 2014).

In this study, we explore the feasibility of a ML concentration forecasting system based on the analogs method. In particular, the surface ML concentration is linked to the atmospheric patterns during a training period. Then, during the forecasting phase, the forecasted atmospheric situation is compared to those realized during the training period to identify analog past situations. The ML concentration during those analog situations is considered to be a good approximation of the ML concentration that will occur during the forecast period. As this is a new approach never tested before for ML dispersion, the first step has been to run several tests to fine-tune the methodology and to characterize its limits of validity. Ideally, the tuning and validation of the method should had been done using in-situ observations but, unfortunately, the available ML concentration datasets are too scarce and this was not possible. Therefore, in this exploratory study, we have used numerically simulated ML concentration fields for the development and validation of the system.

The rest of the paper is organized as follows. In section 2, the statistical method, the datasets used and the different choices tested are introduced. In section 3, the model results are presented and





discussed, and, finally, some conclusions about the capabilities of this new approach are outlined

in section 4.

**2. Data and methods**

**2.1. The analog method**

The implementation of the analog method requires three different sets of data to generate the forecast of the variables of interest. First, we need a dataset of the variables that describe the

atmospheric patterns over the region of study, the so called predictors ($X_i$). The predictors are usually obtained from reanalysis datasets available from meteorological services. The methodology assumes that similar distribution of the predictors leads to similar outcomes of the local variables or predictands ($Y(X_i)$). Therefore, the second dataset consist on spatial patterns of the variable of interest for the same period for which the predictors data is available. These two

datasets constitute the training dataset for the system. The third dataset are the forecasts of the predictors, $X_{ref}$, for the period in which we intend to make the forecast of the variable of interest, $Y_{ref}$. These forecasts can usually be obtained from meteorological services. The forecast of the variable of interest on a specific date, $Y_{ref}$, is carried out by finding analog dates within the training period ($t_{an}$) in which the predictor patterns are similar to the forecasted one ($X(t_{an}) \approx X_{ref}$). Then

the value of the variable of interest is estimated as a function of the predictands corresponding to the analog days selected $Y_t = f(Y(t_{an}))$. The selection of the analog dates can be based on different statistical metrics. The most commonly used is the Euclidean distance or root mean square error difference (RMSED) (Zorita et al., 1995; Cubasch et al., 1996; Gutiérrez et al., 2013), although other metrics based on different statistics can also be used.

For our model, the predictors will be the atmospheric conditions, characterized by the Sea Level Pressure ($SLP$) and the wind speed ($U_{10}$, $V_{10}$). The predictands will be the ML concentration outputs from the modeling system developed by Soto-Navarro et al. (2020) and described in the following section. The simulations cover the period 2003 – 2013, which will be the training period in the implementation of the analog method.  A scheme of the model algorithm is shown in figure

1.



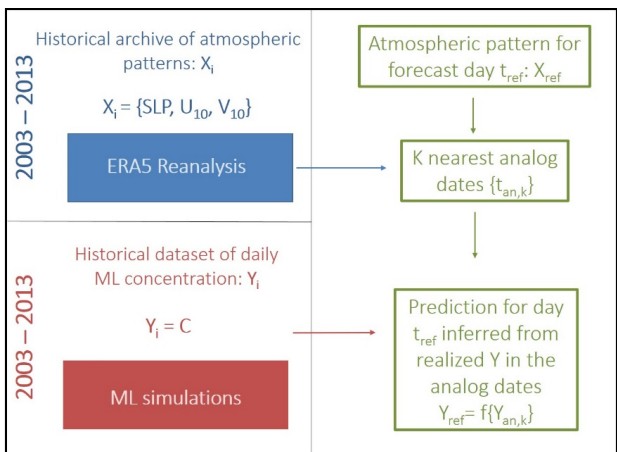

**Figure 1. Scheme of the functioning of the analog method.**

## 2.2. Algorithm implementation

The first step to implement the analog method is to define a cost function $J_M$ that measures the similarity between different meteorological situations. Then, for a given reference day ($t_{ref}$; i.e. the day when the forecast is generated) we estimate how close is the meteorological situation of that day to the rest of the days in the training database by computing $J_M$ for the whole training period. Those days with the lowest $J_M$ values are selected as the analog days ($\{t_{an}\}$). In a second step, the ML concentration maps ($C$) obtained in the training database for those days are combined to produce the analog concentration map ($C_{an}$). In our case we use the median to reduce the influence of extreme concentration values close to ML sources:

$$C_{an} = median(C\{t_{an}\}) \tag{1}$$

$C_{an}$ is therefore considered as the forecasted ML concentration for the reference day.

As mentioned before, there are no suitable observational datasets to validate the system. Homogenized datasets covering a long period of time would be required for this task. Although there are some efforts to develop new databases (Maximenko et al., 2019), up to our knowledge, there are no such datasets in the Mediterranean yet. Thus, in order to have a first assessment of the quality of this approach we can only compare $C_{an}$ with $C_{ref}$ ($= C(t_{ref})$). In other words, we assume the concentration ML maps from the database are the reality and we validate the system against that "virtual reality". This may produce overoptimistic results which will be discussed below.

Several diagnostics are used to characterize the quality of the forecasts. The first one is the root median square error (RMEDSE):





$$RMEDSE = \sqrt{median\left(\left(C_{an} - C_{ref}\right)^2\right)} \tag{2}$$

We have chosen this parameter instead of the root mean square error to reduce the overall impact of outliers linked to very high concentration values close to ML sources. Complementary we also compute the temporal correlation $\rho$:

$$\rho = \frac{Cov(C_{an}, C_{ref})}{\sigma_{C_{an}} \sigma_{C_{ref}}} \tag{3}$$

where **Cov** represents the covariance and $\sigma$ the standard deviation. Additionally, we compute the

RMEDSE ratio (RR) which is defined as the ratio between the RMEDSE of the forecast (eq. 2) and the RMEDSE computed using all the days in the database, RMEDALL:

$$RR = RMEDSE/RMEDALL \tag{4}$$

The lower the value of RR is, the better the forecast is. Values of RR close to 1 means that the quality would be the same than using any random day, so the forecast is not providing any new

information. RMEDSE, $\rho$ and RR are computed spatially and/or temporally.

We have performed an ensemble of experiments to evaluate different options for the configuration of the method. First, we have tested 4 different definitions for the cost function $J_M$:

$$JM_1 = \sqrt{\overline{\left(\left(SLP(t) - SLP(t_{ref})\right)^2\right)}} \tag{5}$$

$$JM_2 = \sqrt{\overline{\left(\left(u_{10}(t) - u_{10}(t_{ref})\right)^2 + \left(v_{10}(t) - v_{10}(t_{ref})\right)^2\right)}} \tag{6}$$

$$JM_3 = JM_1(t)/\langle JM_1(t)\rangle + JM_2(t)/\langle JM_2(t)\rangle \tag{7}$$

$$JM_4 = \sum_{t_{ref}-\Delta t}^{t_{ref}} JM_3(t) \tag{8}$$

So, the similarity between meteorological situations is assessed either in terms of the sea level pressure (**SLP**, **$JM_1$**), the 10-m winds (**$U_{10}$**, **$V_{10}$**; **$JM_2$**), a normalized combination of both (**$JM_3$**) or the cumulated values of **$JM_3$** during a period (**$\Delta t$**) before the reference day (**$JM_4$**). In our case,

**$\Delta t$** has been set to 7 days. Note that the horizontal bars indicate spatial averages for (**$JM_1$**) and (**$JM_2$**), while $\langle\rangle$ in (**$JM_3$**) denotes temporal mean.

In order to test if the method shows different skills depending on the domain of application, we have applied the method to seven different regions: the whole Mediterranean, the eastern and western basins, and in the Gulf of Lions, the region around the Balearic Islands, the Adriatic Sea

and the Aegean Sea (see Figure 2). In each case, the analog days have been defined using only data on the selected region.





Additionally, we have tested if the skill of the method depends on the time scales of the ML concentration variability. So, in addition to use the ML concentration dataset, we have used two filtered versions of it separating those processes above and below 15 days ($C_{hi\text{-}freq}$ and $C_{lo\text{-}freq}$).

Finally, for completeness, we propose three additional models for the forecasting. First, we forecast the concentration change in 7 days ($\Delta_{7d}C$). The underlying idea is that the meteorological situation could be a better predictor of the rate of change than of the absolute value (e.g. winds may determine the changes in the concentration rather than the absolute value). The second one is to simply assume 7-days persistence as the forecasting model (I.e. we assume $C(t_{ref}) = C(t_{ref\text{-}7}$

$_{days})$). This model will tell us if having a good observational characterization of the ML concentration would be a good predictor of what will be the situation one week later. The last one is a combination of the previous two: we add the forecast of the concentration change to the 7 days persistence ($C(t_{ref}) = C(t_{ref\text{-}7\ days}) + \Delta_{7d}C$). In other words, we test if combining a good observational characterization of the ML concentration with an analog-based forecast of the

concentration change can improve the results.

**2.3 Reanalysis data for the atmospheric fields**

The period considered as historical for the training of the analog method is 2003 – 2013, which coincides with the period simulated by the ML dispersion model (as described in the following section). The climatic dataset necessary for the model training and the forecasting is based on the

ERA5 reanalysis dataset, available at the Copernicus Climate Change Service (C3S) web platform (https://climate.copernicus.eu/climate-reanalysis). All the information regarding the ERA5 characteristics can be found on the C3S website.

Two variables have been considered for the characterization atmospheric patterns forcing the ML dispersion in the historical period: the wind speed at 10 meters height ($U_{10}$, $V_{10}$), and the sea level

pressure (SLP). Daily mean values of these variables over the Mediterranean Sea were downloaded and processed for the whole period. The spatial resolution of the atmospheric data is 0.25° (~25 km) and cover the whole Mediterranean basin and the region of the North Atlantic adjacent to the Iberian Peninsula. Figure 2 shows as an example the average SLP for year 2013 in the selected domain.





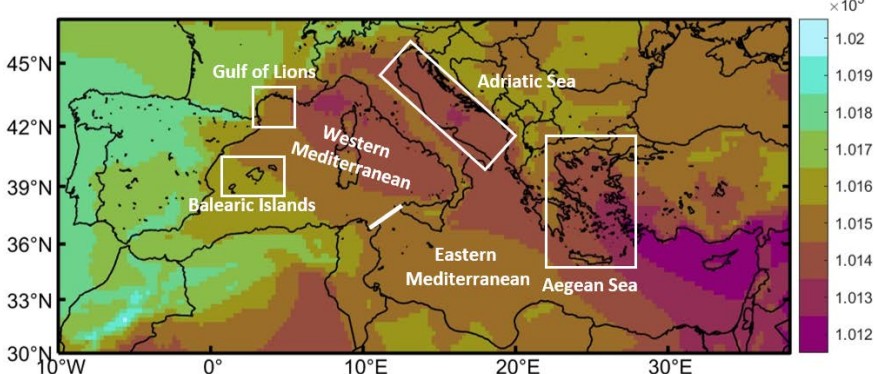


**Figure 2. Average SLP ( in Pa) for the year 2013 in the region, computed from the ERA5 datset. The red line at the Strait of Sicily marks the boundary between the Western and Eastern basins. The red rectangles limit the sub-basins of the Baleric Islands, The Gulf of Lions and the Aegean Sea, where specific analyises were carried out.**

### 2.4 ML concentration data


The ML concentration data is obtained by the simulations performed by Soto-Navarro et al., (2020), as they are considered to be among the most realistic for the Mediterranean Sea. Due to the relevance of the quality of the ML concentration data, details on the modelling system are presented below. The system is based on two components, a regional high resolution circulation model (RCM) reproducing the 3D current velocity field in the Mediterranean (NEMOMED36), and a lagrangian model that simulates the evolution of floating particles (Ichthyop 3.3).


### 2.4.1 Regional circulation model

The hydrodynamical model used to simulate the Mediterranean current field is an implementation on the NEMO model, with a spatial resolution of 1/36 degrees (~ 3 km), 50 vertical *z-levels*, stretched towards the surface, namely NEMOMED36. The model produced daily 3D currents along the period 2003 – 2013. MEDATLAS-II temperature and salinity data from 1958 to 1986 is used for the model spin-up. The domain covers the whole Mediterranean basin and has western open boundary, where an Atlantic buffer zone is use to restore salinity and temperature towards the Levitus et al. (2005) climatology, while the volume conservation is achieved by dumping the sea surface height (SSH) to the outputs of previous simulations which assimilate altimetry data. RivDis dataset (Ludwig et al., 2009) is used to simulate the river discharge, which is computed as a freshwater input in the domain's pixel closer to the river mouths. This includes the Black Sea inflow, computed as a river input located in the Dardanelles Strait. The atmospheric forcing is a dynamical downscaling performed by the APEGE-Climate model using spectral nudging, namely ARPERA. ECMWF fields ([https://climatedataguide.ucar.edu/climate-data/era40](https://climatedataguide.ucar.edu/climate-data/era40)) drive the








spectral nudging above 250 km, while small scales are permitted to freely develop. Note that the forcing of NEMOMED36 (ARPERA) is not the same that the one used to characterize the meteorological situations (ERA5). Although both datasets are very similar, they are not exactly the same, thus mimicking the inaccuracies that atmospheric forecasts will inherently have.

**2.4.2 Lagrangian model**

The Individual Based Model (IBM) Ichthyop 3.3 (http://www.ichthyop.org/) is used to determine the 3D trajectories of the virtual ML particles. Using the NEMOMED36 current field, a fourth order Runge – Kutta integration scheme is applied to solve the movement equations. Applying tri-lineal interpolation in space and linear interpolation in time, the euulerian velocities are

computed in the exact location of the virtual particles. The horizontal diffusivity is simulated by a random-walk added to the trajectories. In the coastlines and the domain's boundaries the configuration of the model is set as "*bouncing*", meaning that the particles rebound back to the sea when reaching coastal pixels or the boundary of the domain. Therefore, no beaching scheme is implemented. The uncertainties inherent to the simulation of the beaching process, particularly

when using an oceanic model which does not resolve coastal processes, has made us decide to choose the bouncing scheme, as the beaching algorithm would likely yield overestimated beaching results (Soto-Navarro et al., 2020). The particles position is computed every 15 minutes, and the outputs are save every 24 hours.

**2.4.3 ML Simulations**

Following the estimations of Jambeck et al., (2015), a total input of 100k tons of plastic per year into the whole Mediterranean Sea is set in the model. This total amount is distributed in three different types of sources: cities, rivers and maritime traffic or ships - lanes, according to the ratio 50:30:20% respectively. The 50k tons of plastic per year corresponding to the cities are redistributed in proportion to their population. The cities have been selected as those with a

population higher than 25k inhabitants (a total of 480), most of them located along the shores of Spain, France and Italy (fig. 3). The 30k tons per year of the rivers are distributed among the fifteen main rivers of the basin in proportion to their mean discharge between 1980 and 2012, estimated by the OCHIDEE River Flow model. The position of the coastal sources (cities and river mouths) is selected as the model ocean grid point closer to its actual location. The 20k tons

corresponding to the shipping lanes are uniformly distributed over the regions where concentrations of higher maritime traffic are (Marine Traffic, 2015).

The modelling period covers ten years, between 2003 and 2013. Due to computational limitations, it has been divided in 120 simulations, each one running one year and starting the first day of each month. A total of 41872 particles are released every month, which for the complete experiment

makes a total of more than 5 million particles. The initial concentrations at the different sources



location are represented in figure 3a. The experiments were carried out using particles with positive (floating), neutral, and negative (sinking) buoyancy. In this study, only the results for floating ML particles have been used. Soto-Navarro et al. (2020) showed that the dispersion patterns for floating and neutral particles are very similar, hence the results described below can

be considered valid also for neutral particles.

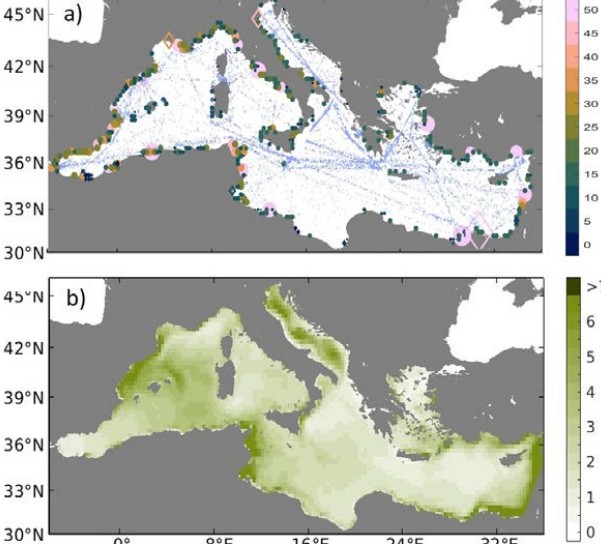

**Figure 3. a) Spatial distribution of initial marine litter concentrations (in kg/km² ) for the three simulations. Circle filled points indicate cities, diamonds indicate rivers and points over the sea indicate the ship lines. b) Average ML concentration of neutral particles (kg/km²) for the period 2003 – 2013.**

The results of the numerical experiments are processed to produce average ML concentration maps over the Mediterranean basin. These maps are computed by dividing the Mediterranean basin in a regular grid of 0.25° x 0.25° cells. The average concentration is estimated as the number of particles in each cell, divided by the cell surface, at each time step. Figure 3b shows the average ML concentration in the Mediterranean for the whole simulated period.

**3. Results**

**3.1 Time variability**

The temporal correlation and the RR of the ML concentration reconstruction using different cost functions and forecasting models are presented in Figures 4 and 5. The spatial patterns of the correlation are very consistent among the different combinations. The fields are relatively patchy

with the highest values in the eastern basin, close to the Turkey coasts, in the Gulf of Gabes, in the west of Sardinia and towards the north of the Balearic Islands. Conversely, the minimum



correlation values are found in the Alboran Sea, the Algerian basin and the Gulf of Lions. The RR maps are very consistent showing lower values where/when the correlation is higher and values closer to 1 where/when the correlation is lower.

Concerning the different cost functions used to identify the analogue situations, the performances using only SLP ($JM_1$) or only wind ($JM_2$) are very similar. Using both variables the quality slightly increases ($JM_3$) and becomes significantly better when using the 7-days average ($JM_4$). For model 1 (forecasting the concentration), the averaged correlation using each cost function is 0.24, 0.25, 0.28 and 0.35 while the averaged RR is 0.93, 0.93, 0.90 and 0.86, respectively. The

forecasting of the concentration change is worse for all cost functions, with averaged correlation values ranging from 0.08 to 0.19 and RR ranging from 1.00 to 0.98. In the light of these results, from now on, we will only consider the results of the analog-based forecast models that use the cost function $JM_4$ (i.e. the one considering the 7-day averaged differences). Using it for forecasting the ML concentration we obtain correlation values ranging from 0.20 to up to 0.60

depending on the region. When forecasting the ML concentration change the values range from non-significant to 0.40 (see Figure 4).

Using 7-days persistence to forecast the ML concentration (Model 3, see Figure 4) the results largely improve. They show correlations that range from 0.20 in the Alboran Sea and the Gulf of Lions to 0.82 around Cyprus, with an average value of 0.60. The RR reaches values as low as 0.4

with an average value of 0.79. Finally, combining both methodologies in Model 4 provides the best results. Combining the 7-days persistence with the analog-based forecast of the concentration change increases the forecasting skills. In this case the averaged correlation is 0.62 and the averaged RR 0.79.




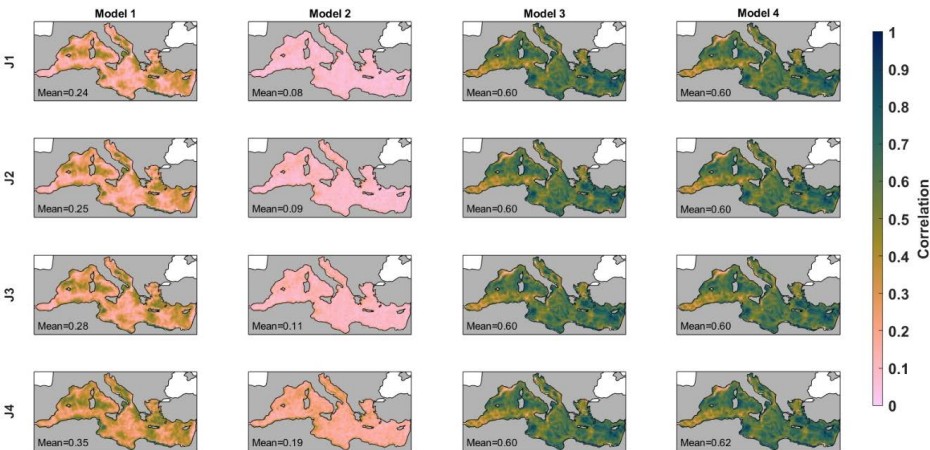

**Figure 4. Temporal correlation of the forecasts using different models and cost functions with the reference dataset. Each column corresponds to a different forecasting model: the analog-based forecast of the concentration (model 1), the analog-based forecast of the concentration changes in 7 days (model 2), the persistence (model 3), and the persistence in combination with the forecast of the concentration change in 7 days (model 4). Each row corresponds to the different cost functions used to identify the analogs (see text for details). Note that all panels in the third column are the same, as in Model 3 no cost function is used.**

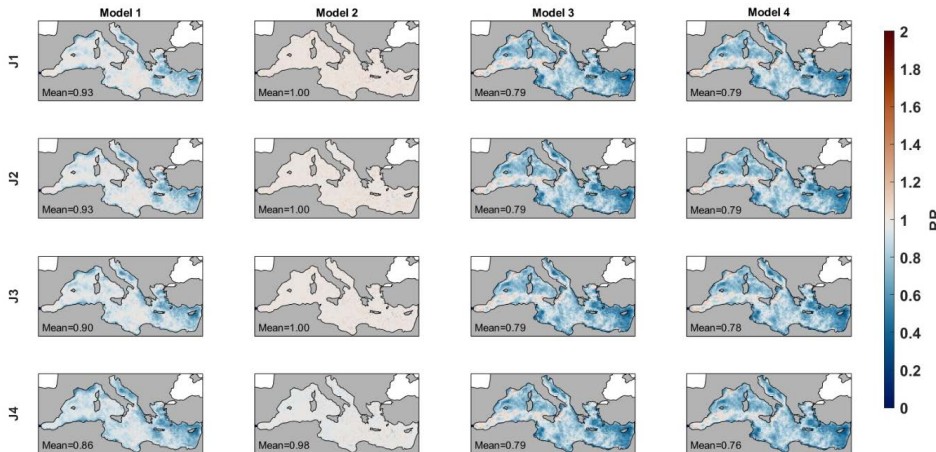

**Figure 5. Same than Figure 5 but for the RMEDSE ratio. Values close to 1 (white) indicate the forecast brings little improvement with respect to use a random day.**

For completeness, we also include an example of the concentration time series for the reference and models 1, 3 and 4 for a point where the forecasts perform well (Figure 6a). It can be seen that Model 1 is well correlated with the reference, showing a good chronology of events although being unable of capturing the concentration peaks. During those periods, the analog-based



forecast largely underestimates the reference values. Models 3 and 4 show almost identical good
       results, as far as persistence is enough to capture most of the variability. The underlying reason
       for this success is that, at this location, the changes of ML concentration are relatively slower, so
       assuming persistence can be a good predictor. For comparison, the time series for a point where
       the models perform poorly are shown in Figure 6b. In this case, the analog-based forecast is

unable to capture any variability and it basically produces the mean value. The other two models
       are able to follow the variability, although in this case the skills are lower than in the previous
       case. The reason is that in this point the ML concentration varies more rapidly, so assuming the
       persistence is not as good predictor as it was in the previous location.

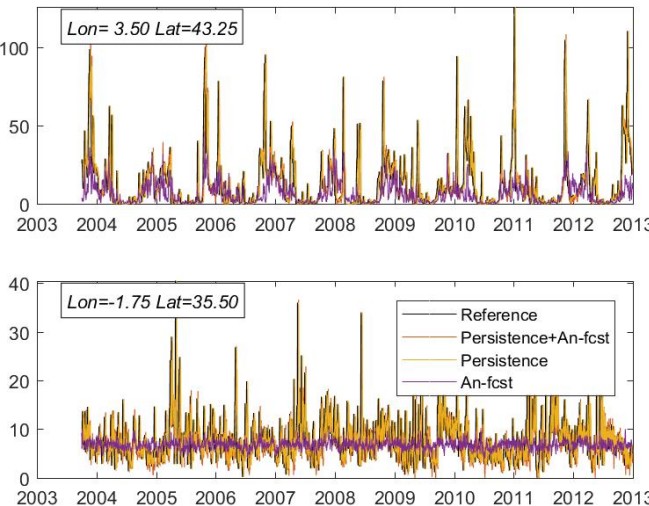

**Figure 6. Time series of ML concentration (in kg/km$^2$) for (top) a location where the analog-based forecast works**
       **well and (bottom) a location where it performs worse. The plots show the reference values, the analog-based**
       **forecast, the persistence and the persistence in combination with the forecast of the concentration change.**

       **3.2 Spatial Variability**

       A complementary view of the performance of the different forecasting models can be obtained

looking at the ML concentration anomalies (i.e. with respect to the temporal mean) in given dates.
       In Figure 7, we show the results for a date when models show good agreement with the reference
       (spatial correlation values are 0.70, 0.76 and 0.78 for models 1, 3 and 4, respectively). All three
       models are able to identify the areas of high and low concentration. Maximum values in the north
       of the Balearic Islands, the Gulf of Gabes and south of Italy and minimum values in the Adriatic

Sea, the Algerian basin and the easternmost part of the Med are well captured. The analog-based
       forecast (Model 1) shows smoother patterns with less low extremes. This is in good agreement
       with what has been seen in the time series in Figure 6, suggesting that this model has difficulties
       to capture very high concentration values. Regarding the persistence-based models, for this



particular date, they perform very well capturing not only the large scale patterns but also the
local features. Looking at a date when the performance is lower something interesting appears.
Although the spatial correlation of Model 1 is not significant (Figure 8b), the large scale features
seem to be well captured. However, the small scale features are clearly not captured which
degrades the spatial correlation. This would also support the previous finding reinforcing the idea
that the analog-based forecast performs better for the large scale features. In places or dates
where/when the small scale features become dominant, the performance of the model drops.

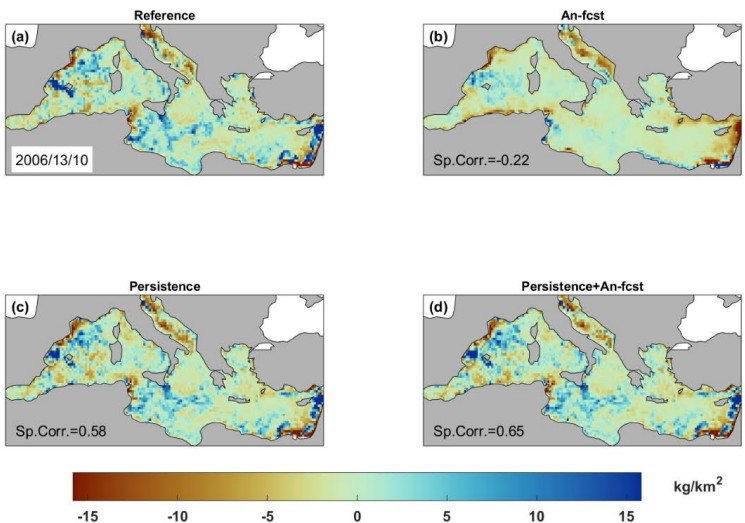

**Figure 7. Maps of ML concentration anomaly for a date where the analog-based forecast performs well (a)
Reference (b) Analog-based forecast (c) Persistence (d) Persistence in combination with the forecast of the
concentration change in 7 days.**



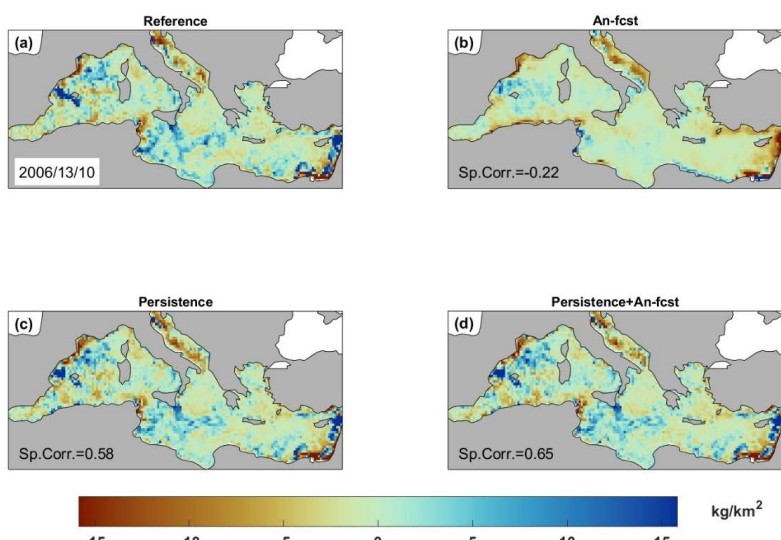

**Figure 8. Like Figure 7 but for a situation where the forecasts perform worse.**

The time series of the spatial correlation and spatial RR at each time step are presented in Figure
9. The results show similar results for the three models forecasting the ML concentration (Models
1, 3 and 4). The skills of the forecasts show a high temporal variability with correlation values
ranging from 0.5 to almost 1 and an averaged value of 0.78, 0.81 and 0.84, respectively. For RR
the values range from 0.3 to more than 1 with an average value of 0.76, 0.79 and 0.71,
respectively. This diagnostic also confirms that the best model is the one combining the
persistence with the forecast of the concentration change.

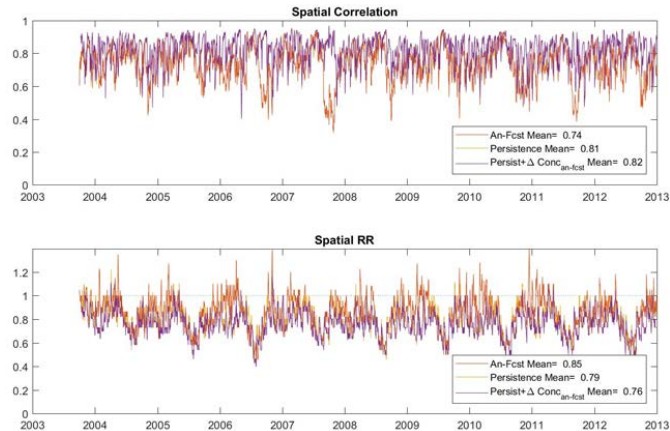

**Figure 9. Time series of (Top) Spatial Correlation and (Bottom) Spatial RR. for the analog-based forecast (Model
1), the persistence (Model 3), and the persistence in combination with the forecast of the concentration change
(Model 3).**





### 3.3 Regional dependence of the forecasting skills

The methodology has also been applied to different domains. That is, the cost function $\mathbf{J_M}$ has

been computed in the regions defined in Figure 2 and the validation has been performed looking only at the ML concentration in those regions. In general, better results are obtained when the analog-based forecasts are applied to a larger region (see Table 1 and Table 2). For instance, the analog-based forecast (Model 1) provides modest results with correlation of 0.31 and 0.35 and RR of 0.92 and 0.86 for the eastern and western Mediterranean, respectively. At local scale the

correlation ranges between 0.29 and 0.34 and the RR ranges between 0.80 and 0.94. The analog-based forecast for the concentration change (Model 2) shows lower skills with correlation below 0.23 and RR above 0.98 in all regions. Both models show better performance forecasting the low-frequency component than the high-frequency one. The correlation of Model 1 forecasts in the different region ranges between 0.31 and 0.40 for the low frequency while it ranges between 0.15

and 0.22 for the high frequency. Consistent results are found when looking at the RR and Model 2 forecasts.

The 7-days persistence (Model 3) shows to be a good predictor for the full signal and the low-frequency component while it struggles to capture the high-frequency variability as expected. Provided that the low-frequency part of the signal is what dominates the ML concentration

variability, this model shows good skills for the full signal with correlation in all regions ranging from 0.55 to 0.64 and RR ranging from 0.75 to 0.82.

The best results for the forecast of the ML concentration are obtained combining the persistence with the analog-based forecast of the 7-days concentration change (Model 4). The averaged temporal correlation is over 0.54 in all regions reaching a value of 0.65 when applied to the

Western Mediterranean, while RR is below 0.80 and reaches 0.76 for the whole Mediterranean.




| Correlation | Full | | | | High-frequency | | | | Low-frequency | | | |
|---|---|---|---|---|---|---|---|---|---|---|---|---|
| | M1 | M2 | M3 | M4 | M1 | M2 | M3 | M4 | M1 | M2 | M3 | M4 |
| Mediterranean | 0,35 | 0,19 | 0,60 | 0,62 | 0,19 | 0,19 | -0,13 | -0,08 | 0,40 | 0,25 | 0,95 | 0,96 |
| East Med | 0,31 | 0,19 | 0,55 | 0,57 | 0,19 | 0,18 | -0,13 | -0,08 | 0,34 | 0,22 | 0,95 | 0,95 |
| West Med | 0,35 | 0,16 | 0,64 | 0,65 | 0,15 | 0,15 | -0,12 | -0,08 | 0,41 | 0,24 | 0,96 | 0,96 |
| Gulf of Lions | 0,29 | 0,20 | 0,51 | 0,54 | 0,20 | 0,19 | -0,16 | -0,10 | 0,31 | 0,19 | 0,94 | 0,94 |
| Balearic Islands | 0,36 | 0,18 | 0,60 | 0,62 | 0,17 | 0,17 | -0,14 | -0,09 | 0,40 | 0,20 | 0,95 | 0,96 |
| Adriatic Sea | 0,31 | 0,23 | 0,53 | 0,56 | 0,22 | 0,21 | -0,13 | -0,07 | 0,28 | 0,17 | 0,94 | 0,94 |
| Aegean Sea | 0,34 | 0,21 | 0,55 | 0,59 | 0,22 | 0,20 | -0,12 | -0,06 | 0,39 | 0,19 | 0,94 | 0,94 |

**Table 1. Regionally averaged temporal correlation of the different forecasting models (M1-M4) applied in different regions (see Figure 2). The models have been applied to forecast the full signal of ML concentration, the high frequency component (period < 15 days) and the low frequency component (period > 15 days).**

| RR | Full | | | | High-frequency | | | | Low-frequency | | | |
|---|---|---|---|---|---|---|---|---|---|---|---|---|
| | M1 | M2 | M3 | M4 | M1 | M2 | M3 | M4 | M1 | M2 | M3 | M4 |
| Mediterranean | 0,86 | 0,98 | 0,79 | 0,76 | 0,97 | 0,98 | 1,47 | 1,44 | 0,82 | 0,92 | 0,26 | 0,24 |
| East Med | 0,92 | 0,98 | 0,82 | 0,80 | 0,98 | 0,98 | 1,48 | 1,45 | 0,89 | 0,94 | 0,28 | 0,27 |
| West Med | 0,86 | 1,00 | 0,75 | 0,74 | 0,98 | 0,99 | 1,47 | 1,45 | 0,79 | 0,93 | 0,25 | 0,23 |
| Gulf of Lions | 0,80 | 0,98 | 0,80 | 0,79 | 0,97 | 0,98 | 1,49 | 1,46 | 0,79 | 0,96 | 0,26 | 0,25 |
| Balearic Islands | 0,94 | 0,98 | 0,76 | 0,75 | 0,98 | 0,99 | 1,48 | 1,46 | 0,91 | 0,96 | 0,27 | 0,25 |
| Adriatic Sea | 0,84 | 0,98 | 0,79 | 0,77 | 0,98 | 0,99 | 1,46 | 1,44 | 0,88 | 0,98 | 0,29 | 0,28 |
| Aegean Sea | 0,85 | 1,00 | 0,80 | 0,75 | 0,97 | 0,99 | 1,46 | 1,45 | 0,82 | 0,97 | 0,27 | 0,26 |

**Table 2. Same as Table 1 but for the RMEDSE ratio.**

The spatial diagnostics have also been computed applying the models to different domains (Table 3). In this case, the analog-based forecast of concentration (M1) show average spatial correlations higher than 0.62 when applied to any region reaching up to 0.94 in the Aegean Sea. Also, the analog-based forecast of concentration change (M2) shows significant average correlations ranging between 0.19 and 0.30. 7-days persistence (M3) is again improving the results although the combination of persistence and the analog –based forecast of concentration change (M4) is the best model when applied in any region. Average correlation ranges between 0.67 and 0.96 and RR is lower than 0.83 everywhere.





|  | Correlation | | | | RR | | | |
|---|---|---|---|---|---|---|---|---|
|  | M1 | M2 | M3 | M4 | M1 | M2 | M3 | M4 |
| Mediterranean | 0,75 | 0,23 | 0,83 | 0,84 | 0,86 | 1,00 | 0,80 | 0,75 |
| East Med | 0,70 | 0,23 | 0,75 | 0,76 | 0,89 | 0,97 | 0,85 | 0,82 |
| West Med | 0,76 | 0,22 | 0,84 | 0,85 | 0,86 | 1,00 | 0,78 | 0,72 |
| Gulf of Lions | 0,65 | 0,19 | 0,66 | 0,67 | 0,83 | 0,98 | 0,85 | 0,83 |
| Balearic Islands | 0,62 | 0,20 | 0,72 | 0,73 | 0,91 | 0,98 | 0,76 | 0,74 |
| Adriatic Sea | 0,64 | 0,21 | 0,67 | 0,69 | 0,85 | 1,00 | 0,74 | 0,73 |
| Aegean Sea | 0,94 | 0,30 | 0,96 | 0,96 | 0,86 | 1,00 | 0,78 | 0,78 |

**Table 3. Temporally averaged regional correlation and RR of the different forecasting models (M1-M4) applied in different regions (see Figure 2).**

It is worth mentioning that we have also tested other options for the cost function like using different temporal averages or using correlation as similarity metrics but no significant differences have been found. Also, we have tried to change the criterion to define the analog days. Instead of identifying as analogs those days with $J_M$ lower than the 1% percentile of the whole $J_M$ time series, we have used less restrictive criteria (5% or 10%). In both cases the results worsened.

**4. Discussion and Conclusions**

The analog-based forecasting technique has been applied to ML concentration for the first time, up to our knowledge. It has proven to be very inexpensive and relatively easy to set-up, so it is an alternative to direct modelling worth to be considered. A key step in the set-up is to select a suitable cost function and the best threshold to identify the analog meteorological situations. In our case, it seems that using integral definitions for the cost function improve the results. In other words, it is better to identify the analog days based on the history of the meteorological situation. Probably, using a different averaging time for each domain would allow increasing the skills of the analog-based model. However, this fine tuning is out of the scope of this paper, as far as there are no suitable observations to validate it, as it will be discussed later.

The quality of the analog-based forecasts depends on the region of application. Our results suggest that the larger the region of application the better, as we get better results for the whole Mediterranean or for the East/West basins than in smaller local areas. A hypothesis for explaining this result is that using the atmospheric situation as a predictor may not be suitable to capture small scale features (e.g. those related to ocean currents or the interaction with coastlines). Further tests including other predictors could be done to refine the method, including ocean currents, for instance.

Another important point is that the method struggles to capture the extreme values as it produces smooth spatio-temporal patterns of ML concentration. Therefore, in locations or regions where



short intense events or small scale features dominate the variability, the method performs worse. This is also one of the reasons why the temporal skills are relatively low (i.e. temporal correlation and RR, see section 3.1). Conversely, if instead of the time variability, what are aimed at are the
spatial structures, the method shows high skills being able to locate relative maximum and minimum (see section 3.2).

We have also shown that persistence is a very good predictor almost everywhere. This is because the ML concentration changes relatively slowly (i.e. the system has a several days memory), at least at the spatial scales solved by the reference dataset. This means that if reliable information
was available (e.g. from a monitoring program), this could be used as a first guess of the ML concentration several days later. Complementary, the analog-based method has also been applied to forecast concentration change. In this case the results were significantly poorer both to capture the time and spatial variability. However, it can be useful to improve the persistence-based forecasts.

Regarding the reliability of the analog based forecasts that could be generated from this reference dataset, its quality would directly depend on the accuracy of the reference dataset. In our case this dataset comes from the outputs of a realistic modelling (Soto-Navarro et al., 2020). However, the model may have some shortcomings as its spatial resolution, beaching parameterization or realism of ML sources. Consequently, the forecasts would be, in the best case, as good as the model
outputs are. Therefore, it would have been better to validate the different forecasting models against actual observations. Unfortunately, the lack of observations with a suitable spatial and temporal coverage prevents from doing it. In the future, it would be worth setting up a monitoring program with enough spatial and temporal resolution that would allow generating a comprehensive enough reference dataset. This dataset could be used to train the analog-based
forecasting system and to validate other existing systems.

In any case, it is worth noting that the validation of the methodology can be considered as robust. For that purpose, it is not required that the reference dataset is an accurate representation of the actual ML concentration. Only the statistics of the ML concentration spatiotemporal evolution has to be reproduced. And in that sense, the model integrates the effects of a realistic atmospheric
forcing and a realistic ocean current field. So, it is expected that the statistics of the ML concentration field is realistic enough. This extent should also be confirmed by a comprehensive observational dataset, at least in certain regions.

In conclusion, the analog-based forecast method presented here has potential to become a suitable cost effective forecasting method for ML concentration. It could be easily implemented in any
region of the world where a realistic reference dataset is available. In those regions where the





large scale ML concentration patterns dominate the variability the method will probably work better than in regions where the variability is dominated by small scale structures.

## 5. Code and data availability

The code and data required to implement the model described in the paper and to reproduce the
results can be publicly accessed at Jordà and Soto-Navarro, (2022). Additionally, the atmospheric fields can be downloaded from the Copernicus portal (https://climate.copernicus.eu/climate-reanalysis).

## 6. Author contribution

Both authors (GJ and JSN) have contributed equally to the design of the study, the coding of the
modelling system, the performance of the simulations, the analysis of the results and the preparation and revision of the manuscript.

## 7. Competing interest

The authors declare that they have no conflict of interest.

## 8. Acknowledgments

Acknowledgements to the EU-Interreg MPAs Plastic Busters Project: preserving serving biodiversity from plastics in Mediterranean Marine Protected Areas, co-financed by the European Regional Development Fund (grant agreement No 4MED17_3.2_M123_027).

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
