# Peer review of "An analogues based forecasting system for Mediterranean marine litter concentration"

_EGUsphere, 2022_

## Author Response (AR1)

Respond to reviewer #1.

Review of "An analogue based forecasting system for Mediterranean marine litter concentration" by Jordab and Soto-Navarro. The authors provided a forecasting model of marine litter abundance for Mediterranean Sea, where marine litter concentrations were forecasted by finding analog dates under the similar SLP and sea surface winds (U10) conditions. This is indeed an interesting idea, but some points especially in the methods section are required to be more clarified before presentation.

We appreciate your effort in reviewing the manuscript. Find below a point by point answer to your concerns:

First, before creating a forecast model using SLP and U10, the authors should justify that SLP and U10 are enough to reproduce the marine litter concentrations in Mediterranean Sea. The simplest way for the justification might be a correlation analysis between marine litter concentration and these metrological data. However, it was likely that cost functions (Eqs. 5–8) spatially averaged over the field were too simple to reproduce spatio-temporal variation of marine litter.

What you are pointing out here is, precisely, the objective of our study. We developed our model starting from an initial hypothesis: SLP and wind speed can be good predictors for the ML concentration distribution and evolution. The base of this hypothesis is that the surface current field, responsible of the ML transport, is strongly influenced by these atmospheric variables. Thus it is justified to assess up to what extent our hypothesis is true as it is explained in the introduction (L92-103).

Then, instead of looking for simple statistical relationships between variables, which are hard to be significant (as the reviewer foresaw), we opted for using the analogues method. In this method there is no need to assume any prior relationship between predictor and predictands, while is non-linear and spatially coherent (L81-83).

In addition, it was unreasonable to use both SLP and U10 concurrently because they are not independent variables.

We agree that the SLP and wind fields are related at large scale trough the geostrophic equation. But this equation implies a derivative (i.e. winds are proportional to SLP gradients) so they are not colinear and it is reasonable to test different combinations of both fields. Moreover, previous studies on statistical downscaling applied on wave forecasting have found that using a combination of SLP and wind fields yield the best reproduction of wave fields (e.g. Wang et al., 2010, Martínez-Asensio et al., 2016). Therefore, our approach has been to follow the same strategy of those authors and try different combinations of SLP and wind fields to define the cost functions. Our results confirm that using only SLP or winds lead to very different performances, while the use of the combination of both provides the best results. These results endorse our strategy.

To be honest, I met difficulties to follow some points in methods section. The authors compared 'Can' with 'Cref' (P. 5, L. 148) to validate their forecast in the 'virtual reality'. 'Can' is a forecasted ML concentration as described just below Eq. (1), while 'Cref' is also the forecast (see P. 4, L.118, a definition for Yref). Please clarify how these two forecast values were defined in this procedure. In this procedure, it seems likely that the training period is the same as the forecast period (2003–

2013). However, it might be reasonable to separate the training and forecast periods such that a forecast was conducted by seeking an analog date in training period. What was the advantage to conduct the training and forecast in the same period?

We recognize that the method explanation was a bit confusing and we thank the reviewer for pointing this out. We have completely rewritten that section to clarify what has been done including to avoid the use of the term "Training period" and changing the subscript "an" by "fcst" to avoid misunderstandings. We hope that the new version of the manuscript is clearer in this sense.

For answering the reviewer concern, Can was the forecasted ML concentration (now this variable is renamed as $C_{fcst}$). This variable is computed as the median of the ML concentration of the analogue days identified in the reference dataset (eq. 5).

Regarding our specific application, there are no suitable observational datasets to validate the forecasting system. Thus, in order to have a first assessment of the quality of this methodology we have to use the concentration ML maps from the database as a "virtual reality" and compare the forecast ($C_{fcst}$) with the C in the database for the forecast date ($C(t_{fcst})$).

To define the forecast day, we pick any date from the reference period and forecast the ML for that day using all the data available, except for a week before and after of the forecast day to avoid spurious good results due to autocorrelation. This has been repeated for all the dates in the reference period (3650) and several statistical quantities have been computed to assess the skills of the method.

With this methodology there is no need of having a "training" period as it is required in many AI algorithms. As we do not use the meteorological fields around the selected forecast date, the quality of the results is not biased high. We have also tried to use only part of the dataset for the training and the rest for the validation but the results are very much alike. So, we have preferred to use the whole period, as it is explained in the manuscript, because the statistics are more robust.

**Specific points**

P.4, L.110, What does the subscript 'i' stand for? Why 'i' disappeared in the following variables?

In this section we make the theoretical description of the analogues methodology. Xi are the i predictors used by the model (which in our case are the SLP, U10 and V10). Later on we don't mention these generic variables anymore. However, we think that the reviewer is right and it can be confusing, so in the new version of the manuscript we have removed the 'i'.

P.4, L. 121, "Yt=f(Y(tan))"; What does the subscript 't' stand for? What was the difference between Yt and Yref? What was the form of the function 'f'?

There was a typo in the formula and should be $Y_{ref}$ instead of $Y_t$. Here '$t_{an}$' refers to the time (date; 't') of the analogue days. The prediction $Y_{ref}$ (now $Y_{fcst}$) is constructed from the fields of the analogue days '$Y(t_{an})$' using a function f, so $Y_{ref}= f(Y(t_{an}))$. These definitions are general, valid for any model based on the analogues theory. Later, when describing our implementation, we substitute the predictand Y by the ML concentration (C; see the new methodology section). Regarding the function 'f', in our case it is simply the median value (see eq.5 and lines 149-160).

P.6, Eq. (5–8); In my understanding, these cost functions were used for seeking the analog date so that the cost functions become the lowest in the time series. To show time series of the cost functions and date chosen for the analog dates help the readers understand the procedures.

The reviewer is right. We have included a figure with the time series of a cost function to illustrate how it works (figure 1b of the new version).

Respond to reviewer #2.

A drawback of this exercise and other ML models is the scarcity of field data. The authors solve this problem by using predictions of ML concentrations from a previously published model based on the current field (Soto-Navarro et al. 2020), referred to as "synthetic reality". The authors make this limitation clear in the abstract, introduction and conclusions, demonstrating great scientific integrity.

Thank you very much for the effort of reviewing the manuscript and for your kind words. Find below a point-by-point answer to your concerns:

Applying the analogues method for the period 2003-2013, the authors are able to reproduce the results derived from the regional circulation model quite well, although I could not differentiate calibration and validation periods (this point could be improved). The main advantage of the new approach lies in the lower computational effort.

We recognize that the method explanation was a bit confusing, and we have completely rewritten that section to clarify what has been done including to avoid the use of the term "Training period" and changing the subscript "an" by "fcst" to avoid misunderstandings.

To define the forecast day, we pick any date from the reference period and forecast the ML for that day using all the data available except for a week before and after of the forecast day, to avoid spurious good results due to autocorrelation (see lines 149 - 160 in the new version). This has been repeated for all the dates in the reference period (3650) and several statistical quantities have been computed to assess the skills of the method.

With this methodology there is no need of having a calibration period as it is required in many AI algorithms. As we do not use the meteorological fields around the selected forecast date, the quality of the results is not biased high. We have tried to use only part of the dataset for the calibration and the rest for the validation but the results are very much alike. So, we have preferred to use the whole period as it is explained in the manuscript because the statistics are more robust.

These good results, however, may be partly due to the use of ML concentrations derived from a current field. Indeed, the current field is closely related to the weather variables used as predictors in the analogues model (W and SLP).

That is precisely our hypothesis: the current field is responsible for the ML transport, so if W and SLP are closely related to the current field, perhaps we could use them as predictors for the ML concentration. The main advantage is that, while there are very few observations of ML concentration and the numerical forecasting of ML is computationally expensive, accurate W and SLP forecasts are routinely provided by meteorological services. Moreover, W and SLP have been used in the past to reconstruct wave fields (e.g. Wang et al., 2010, Martínez-Asensio et al., 2016). Therefore, it was worth testing our hypothesis.

Apart from the surface circulation, litter inputs also play a role on the ML concentrations of the synthetic reality. Litter inputs were calculated from the spatial distribution and economic status of population. Since inputs are independent on the weather conditions in the synthetic reality, I suppose that the spatio-temporal variability induced by litter inputs causes divergences in the weather-based analogues model. This could explain why the analogues model is worse at explaining extreme values of ML, often related to large inputs.

The reviewer is totally right. Even if the surface circulation was perfectly forecasted by the SLP and W fields, it would not be enough to perfectly forecast the ML concentrations, as they depend

on the boundary conditions. Thus it was not obvious at all that using a simple model based on W and SLP could lead to reasonable ML forecasts.

In my opinion, Jordá and Soto-Navarro introduce an interesting new method for the study of ML. More empirical support is still needed to reach a proper development of the method. Real data would allow the use of predictive variables more directly related to ML, such as the surface current field or maritime traffic (now used in the "synthetic reality" model).

We agree with the reviewer. Unfortunately, as we point out in the introduction, there is a large knowledge gap regarding the distribution and temporal variability of the ML sources along the coasts (particularly in developing countries). Also, the observations of ML concentration are too scarce to be usable to train forecasting algorithms. Nevertheless, our aim was to provide the tools that could be used once that information will be available.

---

## Author Response (AR2)

Respond to editor's final review:

Thank you again for your submission of this interesting manuscript to Ocean Science. I am glad that, after a somewhat lengthy reviewing process that was caused partly by the difficulty finding reviewers who were willing to review this manuscript, I can accept your manuscript; pending a few minor technical suggestions.

1. I find the abbreviation ML quite confusing; because it is more typically used for Machine Learning. I can imagine that readers new to the field could mistake the abbreviation. In general, I don't see the advantage of using an abbreviation for a term like 'Marine Litter'; why not spell it out? Sure, the manuscript will become (slightly) longer; but it will also become much more readable.

2. Line 119 in the track-changed manuscript: K is written in normal font and as a capital here; is that correct?

3. Line 234 in the track-changed manuscript: use 'days' instead of dates?

Could you respond to these three points? After that, I will accept the manuscript for publication

Dear editor, following your suggestions we have spell out the ML (marine litter) abbreviation along the text, and corrected the lines 119 and 234. We hope that in after these corrections the article is ready for publication.

Best regards,

Javier Soto-Navarro and Gabriel Jordà.